# Detection of CRISPR–Cas and type I R–M systems in *Klebsiella pneumoniae* of human and animal origins and their relationship to antibiotic resistance and virulence

Xue Li,[1] Ling Wang,[2] Jinghuan Lin,[1] Yingjuan Gu,[1] Zhihua Liu,[3] Jing Hu[2]

**ABSTRACT**  The clustered regularly interspaced short palindromic repeats (CRISPR)–CRISPR-associated protein (Cas) and restriction–modification (R–M) systems are important immune systems in bacteria. Information about the distributions of these two systems in *Klebsiella pneumoniae* from different hosts and their mutual effect on antibiotic resistance and virulence is still limited. In this study, the whole genomes of 520 strains of *K. pneumoniae* from GenBank, including 325 from humans and 195 from animals, were collected for CRISPR–Cas systems and type I R-M systems, virulence genes, antibiotic resistance genes, and multilocus sequence typing detection. The results showed that host origin had no obvious influence on the distributions of the two systems (CRISPR–Cas systems in 29.8% and 24.1%, type I R-M systems in 9.8% and 11.8% of human-origin and animal-origin strains, respectively) in *K. pneumoniae*. Identical spacer sequences from different hosts demonstrated there was a risk of human–animal transmission. All virulence genes (yersiniabactin, colibactin, aerobactin, salmochelin, *rmpADC*, and *rmpA2*) detection rates were higher when only the CRISPR–Cas systems were present but were all reduced when coexisting with type I R-M systems. However, a lower prevalence of most antibiotic-resistance genes was found when the CRISPR–Cas systems were alone, and when type I R-M systems were coexisting, some of the antibiotic resistance gene incidence rates were even lower (quinolones, macrolides, tetracyclines and carbapenems), and some of them were higher instead (aminoglycosides, clindamycins, rifampicin-associated, sulfonamides, methotrexates, beta-lactamases and ultrabroad-spectrum beta-lactamases). The synergistic and opposed effects of the two systems on virulence and antibiotic-resistance genes need further study.

**IMPORTANCE**  *K. pneumoniae* is an important opportunistic pathogen responsible for both human and animal infections, and the emergence of hypervirulent and multi-drug-resistant *K. pneumoniae* has made it difficult to control this pathogen worldwide. Here, we find that CRISPR–Cas and restriction–modification systems, which function as adaptive and innate immune systems in bacteria, have synergistic and opposed effects on virulence and antibiotic resistance genes in *K. pneumoniae*. Moreover, this study provides insights into the distributions of the two systems in *K. pneumoniae* from different hosts, and there is no significant difference in the prevalence of the two systems among *K. pneumoniae* spp. In addition, this study also characterizes the CRISPR arrays of *K. pneumoniae* from different hosts, suggesting that the strains sharing the same spacer sequences have the potential to spread between humans and animals.

**KEYWORDS**  *Klebsiella pneumoniae*, CRISPR–Cas system, R–M system, antibiotic resistance, virulence, multilocus sequence typing

**Peer Reviewers** Hassan M. Al-Tameemi, Basrah University, Basrah, Iraq; Abdelfattah Ahmed Abdelkhalek, Future University in Egypt, Cairo, Egypt

Address correspondence to Jing Hu, hjalzh@smu.edu.cn, or Zhihua Liu, zhihualiu@126.com.

Xue Li and Ling Wang contributed equally to this article. The author order was determined by Jing Hu.

The authors declare no conflict of interest.

See the funding table on p. 12.

*[This article was published on 19 December 2024 with errors in one sentence of the abstract. The sentence was corrected in the current version, posted on 8 January 2025.]*

The clustered regularly interspaced short palindromic repeats (CRISPR)−CRISPR-associated protein (Cas) system is part of the adaptive immune system that defends against invading genomes (1). The CRISPR locus consists of short repeat sequences interspersed with unique spacer sequences that are homologous to sequences of invading DNA, and a set of genes encoding nucleases (*cas* genes) are typically located near the CRISPR locus (2). Spacer sequences mostly originate from previously encountered phages or plasmid genomes or other mobile genetic elements (MGEs) in a linear, time-resolved manner; these sequences undergo insertion and selective elimination in the course of microorganism evolution (3). The CRISPR−Cas system is mainly divided into two classes and six types according to constituent proteins and modes of action (4). In addition to such adaptive immunity, innate immunity R-M systems are major players in the coevolutionary interaction between MGEs and their hosts. R-M systems are classified into four main types based on the specific number and types of enzymes in the system (5).

*Klebsiella pneumoniae* is an opportunistic pathogen that causes serious infections in humans and animals (6). It frequently acquires virulence plasmids and antibiotic resistance genes via the horizontal gene transfer (HGT) of plasmids and MGEs (7), and transfers between humans and animals via the food chain and occupational contact (8). The CRISPR−Cas system that protects against the invasion of foreign genetic elements has been reported in *K. pneumoniae*. The most common CRISPR−Cas system is type I-E, which possesses eight *cas* genes (*cas1*, *cas2*, *cas3*, *cse1*, *cse2*, *cas7*, *cas5*, and *cas6*) and either one or two CRISPR arrays (9). Other CRISPR−Cas systems such as type I-F and type IV are also observed in *K. pneumoniae* (9, 10). Four types R-M systems were also identified in *K. pneumoniae*. Type I R-M systems are predominant and three other types (II/III/IV) that were identified were not as structurally complete and widely distributed as the type I R-M system (11). Type I R-M systems are composed of three subunits encoded by three genes, namely, *hsdR* (where *R* stands for restriction), *hsdM* (where *M* stands for methylation), and *hsdS* (where *S* stands for sequence specificity) (12).

The effect of the CRISPR−Cas systems on antibiotic resistance and virulence varies in different bacteria. Previous studies demonstrated how CRISPR−Cas systems both limited the acquisition of the $bla_{KPC}$-IncF plasmid in *K. pneumoniae* (2) and promoted antibiotic resistance in *Campylobacter jejuni* (13). Meanwhile, CRISPR−Cas systems enable the modulation of biofilm production in *Pseudomonas aeruginosa*, which is an important virulence factor for various pathogenic microorganisms (14). A recent study has shown that the type I R-M system is associated with defense against $bla_{KPC}$ plasmid transport into *Escherichia coli* (12), but the relationship between type I R-M systems and virulence is still unknown. Several studies also demonstrated the synergistic effect of the CRISPR−Cas systems and the R-M systems in bacteria. In *Streptococcus thermophiles*, the type II CRISPR−Cas system interacts with type II R-M systems to increase the overall phage resistance of bacteria (15). In *Enterococcus faecalis*, the CRISPR−Cas systems and R-M systems significantly impact the spread of antibiotic-resistance genes (16). In *Staphylococcus aureus*, the type II CRISPR−Cas system and R-M systems team up to achieve long-term immunity (17). However, whether the two systems act synergistically on antibiotic resistance and virulence in *K. pneumoniae* has not been reported.

Here, we investigated the prevalence of these two systems in a collection of whole genomes of *K. pneumoniae* from different hosts as well as their relationship with virulence and antibiotic resistance. This research will help develop novel strategies for preventing the spread of virulence and antibiotic-resistance genes among *K. pneumoniae* in humans and animals.

## MATERIALS AND METHODS

### Strain collection

All the *K. pneumoniae* genomes that are annotated as "chromosome" or "complete" at the assembly level were retrieved from the National Center for Biotechnology Information genome database as of the last 5 years (2019–2023) (18). The isolation source and isolation time were obtained manually from the details page of each genome. For genomes of repeatedly recorded strains, the one with a higher sequencing quality was taken as applicable or otherwise taken randomly. Finally, a total of 520 sequences of *K. pneumoniae* were downloaded in this study, including 325 from humans and 195 from animals (Data S1). Of these, 159 human strains were isolated from 2007 to 2022, but the isolation time of the remaining 66 strains was not recorded. The 195 animals strains that were isolated from 2010 to 2022 included dogs ($n = 70$), cats ($n = 35$), canis lupus familiaris ($n = 19$), bovines ($n = 15$), horses ($n = 13$), *Pteropus poliocephalus* ($n = 10$), chickens ($n = 9$), and swine ($n = 9$). The whole-genome downloads were saved in FASTA format.

### Identification and analysis of CRISPR−Cas systems

The whole-genome sequences were uploaded to the CRISPR−Cas++ website (19) (https://crisprcas.i2bc.paris-saclay.fr/) to obtain CRISPR−Cas information for each strain (including the CRISPR locus, *cas* gene, and repeat and spacer sequences). The genomes with at least one CRISPR array and one Cas at the same time were defined as CRISPR−Cas system positive (20). CRISPR spacers were visualized using CRISPRstudio software (21). The homology of the spacer sequences in *K. pneumoniae* CRISPR arrays was analyzed by BLAST alignment via GenBank.

### Prediction of the R-M systems in the genomes

Methyltransferase and restriction enzymes in bacterial genomes were identified using Restriction-ModificationFinder (https://cge.food.dtu.dk/services/Restriction-Modi-ficationFinder/) (22). Type I R-M systems encode enzymes capable of both methylating and cleaving (restricting) host and foreign DNA. These systems consist of three host specificity determinant (*hsd*) genes, *hsdR*, *hsdM*, and *hsdS* (23).

### Detection of antibiotic resistance genes, virulence genes, and multilocus sequence typing

Acquired antibiotic resistance genes (aminoglycosides, polymyxins, phosphomycins, quinolones, macrolides, clindamycins, rifampicin-associated resistance genes, sulfona-mides, tetracyclines, tigecyclines, methotrexates, beta-lactamases, ultrabroad-spectrum beta-lactamases, and carbapenems), virulence loci (yersiniabactin, colibactin, aerobactin, salmochelin, *rmpADC*, and *rmpA2*), and chromosomal multilocus sequence typing (MLST) were analyzed using Kleborate (https://github.com/katholt/Kleborate) (24).

### Statistical analyses

SPSS software (version 26.0) was used for data analysis. Chi-square tests or Fisher's exact tests were used to evaluate the association of a set of counts or frequencies among *K. pneumoniae* isolate data. All tests were two-tailed, and all tests with a *P* value of <0.05 were considered significant.

## RESULTS

### CRISPR−Cas systems and type I R-M systems in *K. pneumoniae* strains from different hosts

Among the collected strains, approximately one-third (27.7%, 144/520) contained CRISPR−Cas systems. Ninety-seven (29.8%) isolates from humans and 47 (24.1%) isolates

from animals were positive for the CRISPR−Cas systems. The majority (97.9%, 141/144) of the CRISPR−Cas systems belonged to type I-E (Fig. 1A), and only nine (6.3%, 9/144) were positive for type IV-A3 (Fig. 1B), and there were six (4.2%, 6/144) strains with both types present (Fig. 1C). There was no significant difference in the prevalence of each CRISPR−Cas system and the proportion of the CRISPR−Cas system types among *K. pneumoniae* from different hosts ($P > 0.05$, Table 1).

Type I R-M systems were present in 10.6% (55/520) of the strains, of which 32 (9.8%) were of human origin and 23 (11.8%) were of animal origin. There was no significant difference in the prevalence of the type I R-M systems among *K. pneumoniae* from different hosts ($P > 0.05$, Table 1).

Of the strains (17.9%, 93/520) carrying only the CRISPR−Cas systems, 68 (20.9%) were of human origin and 25 (12.8%) were of animal origin, and there was a significant difference between them ($P < 0.05$, Table 1), while only four strains carried only the type I R-M systems (Fig. 1D). Twenty-nine (8.9%) of human origin and 22 (11.3%) of animal origin carried both systems (Fig. 1E). These demonstrated that the strains with the presence of the type I R-M systems had a high probability of also possessing CRISPR−Cas systems (92.7%, 51/55).

## CRISPR spacers of type I-E CRISPR−Cas systems in *K. pneumoniae* strains from different hosts

A total of 2,683 spacer sequences were found in the genomes of 97 strains of *K. pneumoniae* from humans (a mean of 27.7 spacers per isolate), and 775 unique spacer sequences were screened after removing the same sequences by multiple sequence comparisons. The number of spacers in the CRISPR1 locus from 95 strains varied from 2 to 65, and a mean of 20.1 (1915/95) spacers per isolate was observed. In the CRISPR2 locus from 55 strains, the number of spacers varied from 4 to 35, and a mean of 13.7 (754/55) spacers per isolate was observed. In the CRISPR3 locus from four strains, the number of spacers varied from 3 to 10, with an average of 6 (24/4) spacers per isolate.

The genomes of 44 animal strains of *K. pneumoniae* contained a total of 1,136 spacer sequences (a mean of 25.8 spacers per isolate). Through multiple sequence alignment, 416 unique spacer sequences were identified after removing the repeat sequences. The number of spacers in the CRISPR1 locus varied from 6 to 56 in 18 strains, with a mean of 38.2 (687/18) spacers per isolate. In CRISPR2, the number of spacers varied from 10 to 27 in 28 strains, with an average of 16 (449/28) spacers per isolate. CRISPR1 occurs more frequently in *K. pneumoniae* of human origin than that of animal origin. The average number of spacers whether in CRISPR1 or in CRISPR2 in animal strains was higher than that of human strains.

As depicted in Fig. 2, human- and animal-derived *K. pneumoniae* shared 348 unique spacer sequences. *K. pneumoniae* isolated at different times and regions also had the

**TABLE 1** Distribution of the two systems in *K. pneumoniae* strains from different hosts

| Immune system | Type | Total (*n* = 520) | Human (*n* = 325) | Animal (*n* = 195) | *P* value |
|---|---|---|---|---|---|
| | | *n* (%) | *n* (%) | *n* (%) | |
| CRISPR−Cas | I-E | 135 (26.0) | 93 (28.6) | 42 (21.5) | 0.075 |
| system | IV-A3 | 3 (0.6) | 0 (0.0) | 3 (1.5) | 0.052 |
| | I-E + IV-A3 | 6 (1.2) | 4 (1.2) | 2 (1.0) | 1.000 |
| Total | −[d] | 144 (27.7) | 97 (29.8) | 47 (24.1) | 0.156 |
| R-M system | Type I | 55 (10.6) | 32 (9.8) | 23 (11.8) | 0.484 |
| C only[a] | −[d] | 93 (17.9) | 68 (20.9) | 25 (12.8) | 0.020 |
| R only[b] | −[d] | 4 (0.7) | 3 (0.9) | 1 (0.5) | 1.000 |
| C (+) and R (+)[c] | −[d] | 51 (9.8) | 29 (8.9) | 22 (11.3) | 0.381 |

[a]The strains contained only CRISPR−Cas systems.
[b]The strains contained only the type I R-M systems.
[c]The strains contained both CRISPR−Cas systems and type I R-M systems.
[d]−, Not applicable.

A. Type I-E CRISPR-Cas system in *K. pneumoniae*

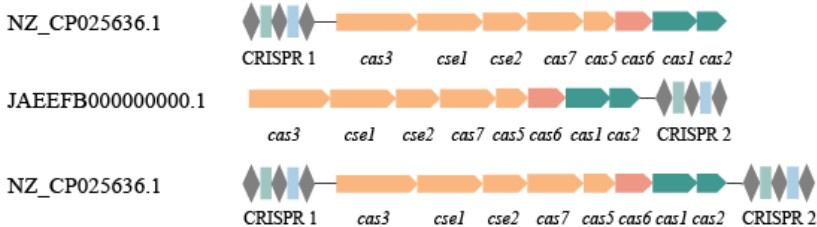

B. Type IV-A3 CRISPR-Cas system in *K. pneumoniae*

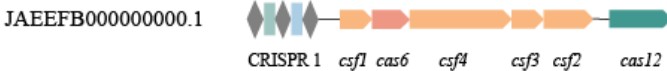

C. Type I-E and type IV-A3 CRISPR-Cas system in *K. pneumoniae*

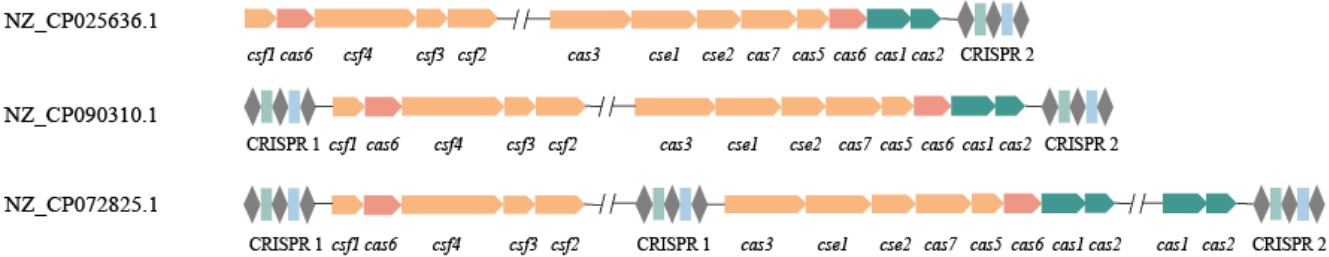

D. Type I R-M systems in *K. pneumoniae*

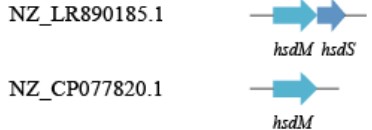

E. Coexistence of CRISPR-Cas systems and type I R-M systems in *K. pneumoniae*

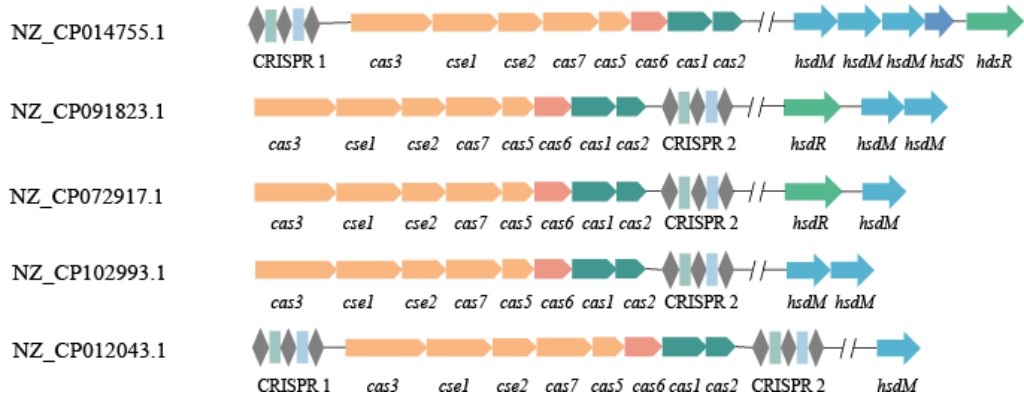

**FIG 1** The structures of CRISPR−Cas systems and type I R-M systems in *K. pneumoniae* strains. Cas genes and CRISPR arrays of CRISPR−Cas systems and genes of type I R-M systems are depicted as arrows in different colors and shapes. The order, orientation, and size of genes and CRISPR arrays were drawn based on the CRISPR−Cas++ website and Restriction-ModificationFinder.

same spacer sequences. Same CRISPR arrays were also found from different hosts, regions, and times. Moreover, most *K. pneumoniae* strains with the same CRISPR arrays had the same sequence types (STs), such as ST15 and ST147.

The homology of the spacer sequences was further analyzed. The results showed that 484 of the 775 spacer sequences (62.5%) of human origin targeted non-self bacterial genomes; 74 (9.5%) targeted the genus *Klebsiella*; 172 (22.2%) were homologous to

phages; and 45 (5.8%) were homologous to plasmids. Similarly, of the 416 spacer sequences of animal origin, 239 (57.5%) targeted non-self bacterial genomes, 58 (13.9%) targeted the genus *Klebsiella*, 84 (20.2%) were homologous to phages, and 35 (8.4%) were homologous to plasmids.

The study clarified the plasmids and phages that were homologous to the spacer sequences of 144 CRISPR−Cas system-positive strains (Data S1), and there were some antibiotic-resistant plasmids such as pKPC2_KP26, pCY-CTXM-15, and pSYCC2_tmex_279 k.

## Relationship among CRISPR−Cas systems, type I R-M systems, and MLST of *K. pneumoniae* strains

MLST analysis revealed that 155 unique STs were identified among the 508 isolates, while 12 strains were not available for MLST information. CRISPR−Cas systems were frequently found in ST15, ST147, ST23, and ST14. Type I R-M systems were commonly found in ST15, ST147, and ST14. On the contrary, CRISPR−Cas systems were rarely found in ST11, ST307, and ST23, and type I R−M systems were also scarce in ST11, ST307, and ST37 (Table 2).

## Relationship of the two systems with virulence and antibiotic resistance in *K. pneumoniae* strains

Because there were only four (0.8%) strains containing only the type I R−M systems, they were not included in the comparison. As shown in Table 3, all three groups of strains had high carriage rates of yersiniabactin, ranging between 43.1% and 62.4%. The carriage rates of all virulence genes in strains with only the CRISPR−Cas systems were all higher than the group with the two systems or group with neither system. However, the addition of the type I R−M systems reduced the positivity of all virulence genes tested in

**TABLE 2** MLST of *K. pneumoniae* isolates with different immunity-related systems[a]

| STs | | | |
|---|---|---|---|
| Number of strains (%) | | | |
| Type I R−M systems present (*n* = 55) | Type I R−M systems absent (*n* = 453) | CRISPR−Cas systems present (*n* = 143) | CRISPR−Cas systems absent (*n* = 365) |
| ST15 | ST11 | ST15 | ST11 |
| 16 (29.0) | 94 (20.2) | 28 (19.4) | 93 (24.7) |
| ST147 | ST307 | ST147 | ST307 |
| 13 (23.6) | 24 (5.1) | 15 (10.4) | 23 (6.1) |
| ST14 | ST23 | ST23 | ST37 |
| 9 (16.3) | 15 (3.2) | 13 (9.0) | 15 (3.2) |
| ST133 | ST37 | ST14 | ST29 |
| 2 (3.6) | 15 (3.2) | 10 (6.9) | 12 (3.1) |
| ST11 | ST15 | ST35 | ST258 |
| 1 (1.8) | 13 (2.7) | 9 (6.2) | 10 (2.6) |
| Others | ST29 | ST45 | ST231 |
| 8 (14.5) | 13 (2.7) | 9 (6.2) | 7 (1.8) |
| | ST258 | ST392 | ST4850 |
| | 10 (2.1) | 7 (4.8) | 7 (1.8) |
| | ST35 | ST485 | ST896 |
| | 9 (1.9) | 3 (2.0) | 7 (1.8) |
| | ST45 | ST11 | ST405 |
| | 9 (1.9) | 2 (1.3) | 6 (1.5) |
| | ST147 | Others | ST147 |
| | 7 (1.5) | 47 (32.9) | 5 (1.3) |
| | Others | | Others |
| | 244 (53.9) | | 180 (49.3) |

[a]ST, multilocus sequence type.

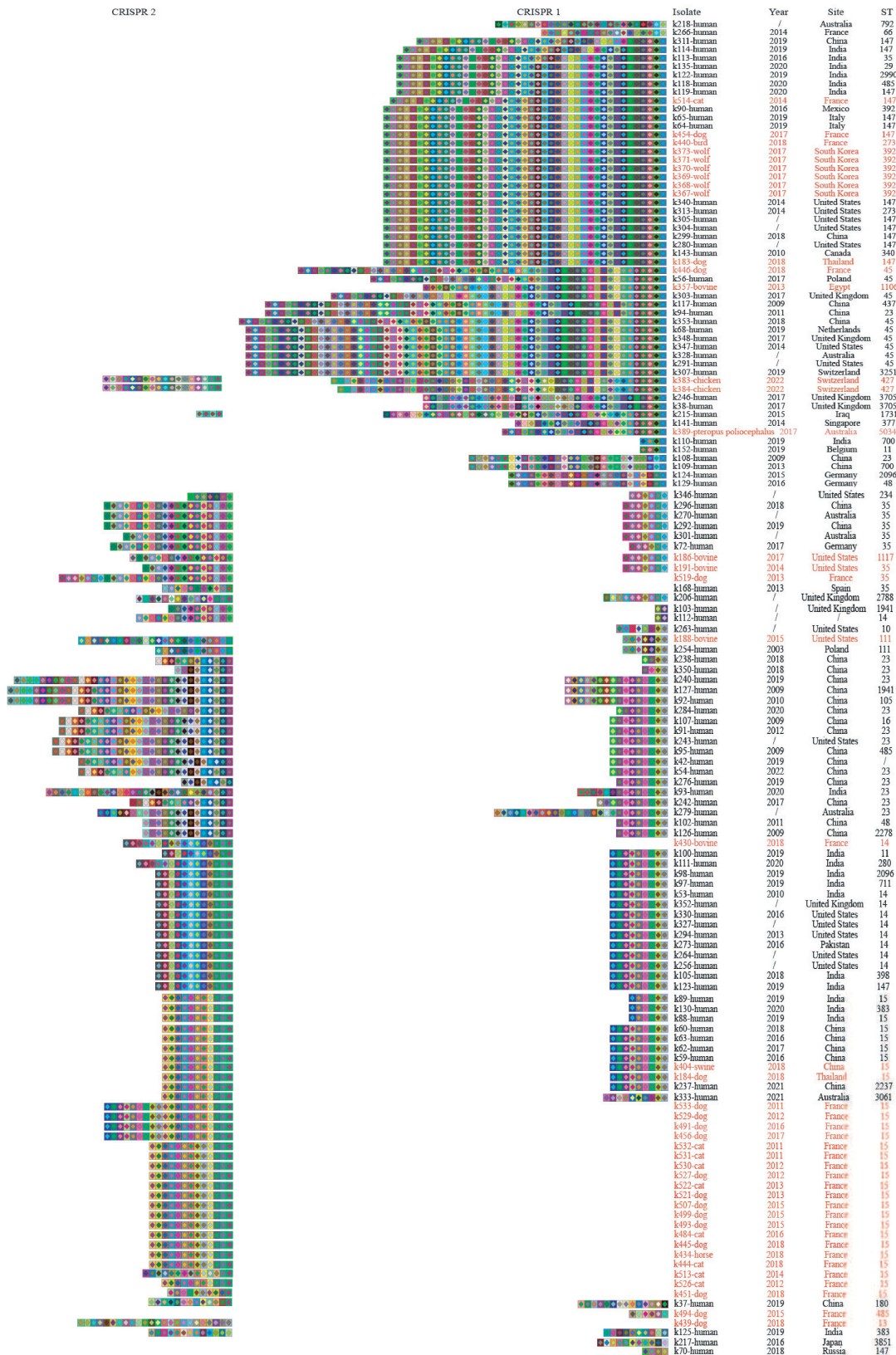

**FIG 2** Graphic illustration of spacer content of CRISPR alleles in 141 *K. pneumoniae* strains. Three CRISPR loci (CRISPR1, CRISPR2, and CRISPR3) in type I-E CRISPR−Cas systems were found in *K. pneumoniae* strains from different hosts, and only four strains had CRISPR3 and were therefore not presented in the figure. Information about animal-origin *K. pneumoniae* strains is labeled in red, and information about human-origin *K. pneumoniae* strains is labeled in black. The (Continued on next page)

Fig 2 (Continued)

repeats have been eliminated, and only the spacers are shown. Each unique spacer is represented by a unique combination of the background color and the color of a particular character based on the software CRISPRStudio. The newly acquired spacer is displayed on the left side, while the earliest acquired spacer is on the right side. ST, multilocus sequence type.

our study, especially for yersiniabactin and colibactin ($P < 0.05$). There was no difference in the carriage rates of all virulence genes between the group with both systems and the group without the two systems ($P > 0.05$).

The study analyzed a total of 14 classes of antibiotic resistance genes. Except for tetracyclines, methotrexates, and ultrabroad-spectrum beta-lactamase resistance genes, all the other antibiotic resistance gene detection rates were higher in the strains lacking both systems than in those with only the CRISPR–Cas systems. This pattern was opposite to that observed with virulence genes (Table 3). When the two systems coexisted, seven classes of antibiotic resistance genes (aminoglycosides, clindamycins, rifampicin-associated resistance genes, sulfonamides, methotrexates, beta-lactamases, and ultrabroad-spectrum beta-lactamases) carriage rates were increased compared with the group with only the CRISPR–Cas systems, especially for aminoglycosides and beta-lactamases ($P < 0.05$), whereas four classes (quinolones, macrolides, tetracyclines, and carbapenems) were reduced, especially for tetracyclines ($P < 0.05$). Overall, the tetracyclines and carbapenem carriage rates of strains with both systems were significantly lower than those of the other two groups ($P < 0.05$). However, for aminoglycoside and beta-lactamase resistance genes, the carriage rates of strains with both systems were significantly higher than those of the other two groups ($P < 0.05$).

## DISCUSSION

*K. pneumoniae* is part of the *Enterobacteriaceae* family and is widely present in the gastrointestinal tract of humans and animals (25). The increasing occurrence of both virulent and multidrug-resistant (MDR) isolates has led the World Health Organization to consider *K. pneumoniae* a major global concern (7). Fortunately, the presence of CRISPR–Cas systems and the type I R–M systems can help *K. pneumoniae* defend against foreign invasion, which may impede the transmission of virulent and antibiotic-resistant plasmids (11, 26). In our study, the prevalence of CRISPR–Cas systems in *K. pneumoniae* was 27.7%, which was lower than that of *Staphylococcus* (44.6%) (27) and *Enterococcus* (35.5%) (28), and it was reported to be different in *K. pneumoniae*, varying from 11.5% to 54.4% (9, 10). The databases now have more details about sequenced strains since clinical researchers have been paying greater attention to *K. pneumoniae* in recent years, so it is understandable that prevalence rates span a wider range. Previous study has shown that part of the CRISPR–Cas systems in the genome of the laboratory strain *Sulfolobus solfataricus* P2A was lost in the absence of invasive genetic elements, suggesting that the CRISPR–Cas systems may become an unnecessary burden for bacteria when bacteria are under pressure beyond the invasion of MGEs, and there may be a tendency to actively lose the systems (29). The presence of CRISPR–Cas systems in the *K. pneumoniae* genome was lower than the average carrier rate of bacteria (45.0%) (30), which may be related to the loss of the CRISPR–Cas systems in *K. pneumoniae* under strong selective pressure for virulence or antibiotic resistance.

Moreover, type I R–M systems were present in 10.6% of *K. pneumoniae*, which is lower than results reported in *Escherichia coli* (37.1%) (12). Interestingly, in this study, we found that the type I R–M systems were present in *K. pneumoniae*, which had a high probability of also having CRISPR–Cas systems. Previous studies demonstrated that genomes encoding R–M systems were more likely to encode CRISPR–Cas systems for both large and small genomes (5). Once the invading MGEs escape from the innate immunity of the type I R–M systems, organisms may acquire resistance to these infectious elements through CRISPR–Cas systems.

**TABLE 3** Virulence and antibiotic resistance genes in *K. pneumoniae* strains

| Genes | C (+) and R (+)[a] (n = 51) | C only[b] (n = 93) | C (−) and R (−)[c] (n = 372) | P value | | |
|---|---|---|---|---|---|---|
| | | | | C (+) and R (+) vs C only | C only vs C (−) and R (−) | C (+) and R (+) vs C (−) and R (−) |
| Virulence genes, n (%) | | | | | | |
| Yersiniabactin | 22 (43.1) | 58 (62.4) | 175 (47.0) | 0.026[e] | **0.008** | 0.655 |
| Colibactin | 0 (0.0) | 16 (17.2) | 9 (2.4) | **0.002** | **0.000** | 0.608 |
| Aerobactin | 1 (2.0) | 2 (2.2) | 5 (1.3) | 1.000 | 0.631 | 0.540 |
| Salmochelin | 0 (0.0) | 4 (4.3) | 1 (0.3) | 0.297 | **0.006** | 1.000 |
| rmpADC | 0 (0.0) | 5 (5.4) | 1 (0.3) | 0.161 | **0.001** | 1.000 |
| rmpA2 | 1 (2.0) | 2 (2.2) | 3 (0.8) | 1.000 | 0.262 | 0.403 |
| Antibiotic resistance genes, n (%) | | | | | | |
| Aminoglycosides | 28 (54.9) | 21 (22.6) | 147 (39.5) | **0.000** | **0.002** | **0.036** |
| Polymyxins | 0 (0.0) | 0 (0.0) | 1 (0.3) | —[d] | 1.000 | 1.000 |
| Phosphomycins | 0 (0.0) | 0 (0.0) | 0 (0.0) | —[d] | —[d] | —[d] |
| Quinolones | 5 (9.8) | 10 (10.8) | 72 (19.4) | 1.000 | 0.052 | 0.121 |
| Macrolides | 2 (3.9) | 10 (10.8) | 46 (12.4) | 0.214 | 0.669 | 0.097 |
| Clindamycins | 14 (27.5) | 15 (16.1) | 67 (18.0) | 0.105 | 0.670 | 0.108 |
| Rifampicin-associated resistance genes | 2 (3.9) | 1 (1.1) | 40 (10.8) | 0.286 | **0.002** | 0.207 |
| Sulfonamides | 15 (29.4) | 18 (19.4) | 120 (32.3) | 0.170 | **0.015** | 0.683 |
| Tetracyclines | 2 (3.9) | 17 (18.3) | 65 (17.5) | **0.019** | 0.855 | **0.013** |
| Tigecyclines | 0 (0.0) | 0 (0.0) | 0 (0.0) | —[d] | —[d] | —[d] |
| Methotrexates | 18 (35.3) | 19 (20.4) | 56 (15.1) | 0.051 | 0.207 | **0.000** |
| Beta-lactamases | 20 (39.2) | 21 (22.6) | 96 (25.8) | **0.034** | 0.521 | **0.044** |
| Ultrabroad-spectrum beta-lactamases | 16 (31.4) | 19 (20.4) | 71 (19.1) | 0.143 | 0.769 | **0.042** |
| Carbapenems | 0 (0.0) | 4 (4.3) | 31 (8.3) | 0.297 | 0.270 | **0.038** |

[a]The strains contained both CRISPR–Cas systems and type I R-M systems.
[b]The strains contained only CRISPR–Cas systems.
[c]Neither system existed in the strains.
[d]−, Not applicable.
[e]Bold signifies P < 0.05.

In *K. pneumoniae*, three types of CRISPR−Cas systems have been identified, including type I-E, type I-F, and type IV-A. Type I CRISPR−Cas systems are mainly present in chromosomes, whereas type IV systems are only found in plasmids (26). In our study, the type IV-A CRISPR−Cas systems were found in chromosomes. Type IV-A CRISPR−Cas systems encode a DinG helicase (c*sf4*) and an effector protein (*cas6*) and lack apparent adaptation modules (*cas1* and *cas2*). Evidence was found that the type IV-A CRISPR−Cas system was interacting with the type I-E CRISPR−Cas system, thus providing a simple answer to the enigmatic absence of type IV adaptation modules (31). It was anticipated that obtaining CRISPR−Cas-positive plasmids would enhance the capabilities of the chromosomal CRISPR−Cas system, providing a wider spectrum of immunity against MGEs (32).

In our study, the CRISPR−Cas systems that coexisted with the type I R−M systems were only type I-E, but the composition of CRISPR loci was diverse. Type I R−M systems were also pleomorphic in *K. pneumoniae* (11), with all type I R−M systems having *hsdM* units and the *hsdR* and *hsdS* units being randomly possessed. The loss of *hsdR* implied that the role of the type I R−M systems was to protect organisms rather than attack MGEs, which was related to the immunocompromised state of strains.

The prevalence of the two systems in *K. pneumoniae* from different hosts was not significantly different, suggesting that host origins had no vital influence on the distribution of the two systems to some extent. A possible reason is that more than half of the animal isolates (53.8%) collected in this study were from companion animals, which had more contact with people and were infected with *K. pneumoniae* by close affinity. For example, ST15 strains had a predominantly companion animal origin and were more consistent with CRISPR arrays of human isolates.

The two CRISPR loci (CRISPR1 and CRISPR2) in type I-E CRISPR−Cas systems were found to be widely distributed in *K. pneumoniae* strains from different hosts. The spacer sequences were thought to serve as a memory bank for CRISPR−Cas systems to resist foreign nucleic acid invasion, and the number of spacer sequences within a locus can reflect the number of invasions (33). The high variability of spacers in both CRISPR1 and CRISPR2 indicated their frequent gains and losses in *K. pneumoniae*. In CRISPR1, the average number of spacers was significantly higher in the animal strains than in the human strains. A similar situation was observed for CRISPR2. In clinical practice, multiple antibiotics are often used in combination to treat a variety of infections, leading to increased selection pressure for bacterial antibiotics, the development of resistance, and susceptibility to invasion by foreign substances, with more spacer sequences in play. In contrast, although some findings suggest that *K. pneumoniae* can cause sickness in a variety of animals (34), *K. pneumoniae* has long been thought to pose little harm to animals. Hence, bacterial infections in animals tend to be underappreciated, with relatively little exposure to antibiotics, which means the bacteria do not have to acquire antibiotic-resistance genes to survive, with less spacer sequences in play. These speculations disagreed with our results, which may be due to the fact that the animal strains analyzed in our study were mostly MDR strains.

In addition, identical spacer sequences were observed in *K. pneumoniae* strains from different hosts isolated at different times and locations, suggesting that these strains may have been invaded by several MGEs showing high relatedness or have undergone HGT. Human isolates were temporally and geographically unrelated to the animal isolates, but the ancestral spacers located at the end of the locus were common, implying that they derived from a common ancestor. This evolutionary trajectory reflected by spacer sequences provides an insight into the transmission of different *K. pneumoniae* genotypes from different sources, regions, and times.

In this study, the proportion of phages and plasmids in *K. pneumoniae*, whether from humans or from animals, was higher than that reported in *Staphylococcus* (27) and *Enterococcus* (28). The higher proportion of spacer sequences of *K. pneumoniae* matching MGEs indicated that *K. pneumoniae* was subjected to more plasmids and phage invasion and that the CRISPR−Cas system was more active. Also, we found that there were some

antibiotic-resistant plasmids that were homologous to the spacer sequences, such as pKPC2_KP26, pCY-CTXM-15, and pSYCC2_tmex_279k, and strains carrying the CRISPR−Cas system with spacer sequences homologous to the plasmids encoding carbapenem resistance were indeed not examined for resistance genes. These indicated that the CRISPR−Cas systems in *K. pneumoniae* probably played a role in regulating HGT.

Moreover, strains with concordant spacer sequences tend to have the same MLST, reflecting the consistency of the two structures in typing. MLST is a nucleotide sequence-based and frequently used typing method for *K. pneumoniae* (35). ST11 is a common MDR ST that is mainly found in Asia and South America and is the predominant type in hospitals and animals in China (36). ST11 strains largely did not carry CRISPR−Cas systems and type I R–M systems, as well as ST307 strains. In contrast, the two systems were widely distributed among ST15, ST147, and ST14 strains, which are important causes of carbapenem-resistant infections, with a reduction in carbapenemase resistance gene acquisition, consistent with former studies (9, 37). ST15 and ST147 strains also crossed over between human and animal origins, implying that there was a risk of human–animal transmission of strains of these two molecular genotypes.

Virulence factors may be encoded by genes in the core genome and in the accessory genome (yersiniabactin locus *ybt*, colibactin locus *clb*, salmochelin locus *iro*, aerobactin locus *iuc*, and regulators of mucoid phenotypes *rmpADC/A2*), and some of the latter genes are harbored on MGEs (7). The yersiniabactin (*ybt*)-encoding ICE*Kp* strongly influences the pathogenicity of *K. pneumoniae* strains (38). In our study, the virulence gene detection rates of the strains carrying only the CRISPR−Cas systems were all higher than those of the other two groups of strains, whereas the addition of the type I R–M systems significantly reduced the detection rates of virulence genes (yersiniabactin and colibactin) in *K. pneumoniae*. The CRISPR−Cas systems in *K. pneumoniae* were not resistant to virulence genes but instead coexisted with them, which was consistent with other reports (26, 39). Fortunately, the addition of the type I R–M systems led to a decrease in virulence gene carriage in *K. pneumoniae*. Future research will concentrate on improving our understanding of the mechanisms.

For the results of antibiotic resistance gene detection, most of the antibiotic resistance gene detection rates were lower in the strains with only the CRISPR−Cas systems than with neither system, which suggested that most antibiotic resistance genes in *K. pneumoniae* were negatively correlated with CRISPR−Cas systems. When both systems were present, the tetracycline and carbapenem resistance gene detection rates of strains were significantly lower than those of the other two groups. Previous research has shown that type I R–M systems may impact the acquisition of $bla_{KPC}$ genes in *K. pneumoniae* (11), and the scarcity of the CRISPR−Cas system was one of the potential factors leading to the propagation of $bla_{KPC}$-IncF plasmids in CG258 *K. pneumoniae* (2). Another prior work also demonstrated that eliminating the CRISPR−Cas cassette in carbapenem-resistant *K. pneumoniae* strains boosted the transformation success of $bla_{KPC}$ plasmids (10). Moreover, in our study, we also found that the carriage of carbapenem resistance genes was greatly reduced in strains when both systems coexisted, suggesting a synergistic effect of the two systems on the antagonism of carbapenem resistance genes.

In this study, we first found a lower detection rate of tetracycline resistance genes in *K. pneumoniae* carrying both systems. Tetracycline antibiotics are a class of broad-spectrum antibiotics that act mainly by preventing the synthesis of bacterial proteins, and the resistance genes are distributed in approximately 130 Gram-positive and Gram-negative genera and are widespread in humans, animals, and the environment (40). Tetracycline antibiotic resistance is usually acquired by HGT. The plasmid-borne *tet*(A) gene is widespread in carbapenemase-producing *K. pneumoniae*, and mutations in *tet*(A) are the main cause of tigecycline resistance (41). Perhaps these two systems could reduce resistance to tetracycline antibiotics in *K. pneumoniae* by blocking the transfer of the *tet*(A) plasmid. Our findings could provide ideas for preventing and controlling the spread of tetracycline resistance genes in *K. pneumoniae*.

In contrast, the detection rates of aminoglycoside and beta-lactamase resistance genes in strains with both systems were significantly higher than the other two groups. Aminoglycosides are important options that can lower the mortality rate effectively in combination therapy with beta-lactam agents (42). The higher resistance percentages for beta-lactams and aminoglycoside resistance genes would become a great challenge for antimicrobial chemotherapy. A study in Egypt reported that the same or higher resistance percentages were found for all beta-lactams among the CRISPR−Cas negative in *K. pneumoniae* isolates compared to others (32). Similarly, another study in China found that isolates carrying the CRISPR−Cas systems had a lower resistance to beta-lactams (43), which contradicted our results. Variable results were found for aminoglycoside resistance genes among the CRISPR−Cas-positive and CRISPR−Cas-negative strains (32, 43). Unfortunately, the addition of the type I R–M systems led to an increase in the two antibiotic resistance gene carriage in *K. pneumoniae*. We supposed that the efficiency of the antibiotic resistance gene acquisition depended not only on the CRISPR arrays and the subunits of type I R–M systems in *K. pneumoniae* but also on the types of antibiotic-resistant plasmids, and there is the possibility of additional regulatory circuits that aid this bacterium in adapting to stress and coordinating between the two systems. Further study is needed to verify the specific impact mechanisms.

However, a limitation of our research is that the data were collected from a variety of sources, and the sources frequently failed to mention the clinical and epidemiological contexts in which the isolates were obtained and the sequences were deposited. Overall, this study provides insights into the distributions of CRISPR−Cas systems and type I R–M systems in *K. pneumoniae* from different hosts and their potential relationship with virulence and antibiotic resistance; these insights may be very helpful in the future to control the public threat of antibiotic-resistant and hypervirulent *K. pneumoniae*. Further research is needed to fully understand the mechanisms involved and to explore the potential of novel strategies to combat *K. pneumoniae* infections.

## ACKNOWLEDGMENTS

This work was supported by the National Key Research and Development Program of China (no. 2022YFC2303600).

J.H. and L.W. conceived and designed the study. X.L. analyzed the data and wrote the paper. J.L. and Y.G. helped collect and analyze some data. J.H., Z.L., and L.W. revised the manuscript. All authors read and approved the final manuscript.

## AUTHOR AFFILIATIONS

[1]Department of Laboratory Medicine, Zhujiang Hospital, Southern Medical University, Guangzhou, Guangdong, China
[2]Department of Nosocomial Infection Administration, Zhujiang Hospital, Southern Medical University, Guangzhou, Guangdong, China
[3]Department of Infectious Disease, Nanfang Hospital, Southern Medical University, Guangzhou, Guangdong, China

## AUTHOR ORCIDs

Zhihua Liu http://orcid.org/0000-0003-4398-4233
Jing Hu http://orcid.org/0000-0002-3696-8115

## FUNDING

| Funder | Grant(s) | Author(s) |
| --- | --- | --- |
| MOST \| National Key Research and Development Program of China (NKPs) | No.2022YFC2303600 | Zhihua Liu |

## ADDITIONAL FILES

The following material is available online.

### Supplemental Material

**Supplemental data (Spectrum00009-24-s0001.xlsx).** 520 *Klebsiella pneumoniae* strains.
**Table S1 (Spectrum00009-24-s0002.docx).** Plasmids and phages homologous to the spacer sequences of 144 CRISPR-Cas system positive strains.

### Open Peer Review

**PEER REVIEW HISTORY (review-history.pdf).** An accounting of the reviewer comments and feedback.

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
