## [Reviewer comments · Microbiology Spectrum]

Microbiology Spectrum

Detection of CRISPR–Cas and type I R-M systems in *Klebsiella pneumoniae* of human and animal origin and their relationship to antibiotic resistance and virulence

Xue Li, Ling Wang, Jinghuan Lin, Yingjuan Gu, Zhihua Liu, and Jing Hu

Corresponding Author(s): Jing Hu, Zhujiang Hospital

Review Timeline:

Submission Date:	January 9, 2024
Editorial Decision:	May 28, 2024
Revision Received:	October 12, 2024
Accepted:	October 31, 2024

Editor: Bobby Warren

Reviewer(s): Disclosure of reviewer identity is with reference to reviewer comments included in decision letter(s). The following individuals involved in review of your submission have agreed to reveal their identity: abdelfattah ahmed abdelkhalek (Reviewer #2); Hassan M Al-Tameemi (Reviewer #3)

Transaction Report:

DOI: <https://doi.org/10.1128/spectrum.00009-24>

Re: Spectrum00009-24 (Detection of CRISPR–Cas and type I R-M systems in *Klebsiella pneumoniae* of human and animal origin and their relationship to antibiotic resistance and virulence)

Dear Dr. Jing Hu:

Thank you for the privilege of reviewing your work. Below you will find my comments, instructions from the Spectrum editorial office, and the reviewer comments.

Revision Guidelines

Sincerely,
Bobby Warren
Editor
Microbiology Spectrum

Reviewer #2 (Comments for the Author):

Good work and great efforts from authors

But the isolates from 2010 to 2022 included dogs It takes a very long time to fix the same conditions, so what are the factors and parameters that help you complete the study at the same conditions?

Reviewer #3 (Comments for the Author):

Detection of CRISPR–Cas and type I R-M systems in *Klebsiella pneumoniae* of human and animal origin and their relationship to antibiotic resistance and virulence

The study collected whole genomes of 520 strains of *K. pneumoniae* from GenBank, including 325 from humans and 195 from animals, for CRISPR-Cas systems and type I R-M systems, virulence genes, antibiotic resistance genes, and multilocus sequence typing detection. Results showed no significant influence of host origin on the distributions of these systems in *K. pneumoniae*. However, there was a risk of human-animal transmission. Virulence gene detection rates were higher when only CRISPR-Cas systems were present but reduced when coexisting with type I R-M systems. The study suggests further study on the synergistic and opposed effects of the two systems on virulence and antibiotic resistance genes. The work is well thought of and very informative. I enjoyed reading this work.

Comments:

1) Plasmids or phages typically encode anti-restriction proteins to enhance their ability to enter a new bacterial host and circumvent the restriction and modification (RM) system. For instance, the plasmid of carbapenemase-resistant *Klebsiella pneumoniae* could encode the anti-restriction proteins ArdA, KlcA, and their homologues. Lines 173-202, 385-388, etc.: The authors did not indicate or discuss the type of plasmids or phages found in the CRISPR sequence analysis. If possible, this could have explained some of the findings, such as the prevalence of carbapenem resistance or the potential for some phage resistance mechanisms.

Mackow NA, Shen J, Adnan M, Khan AS, Fries BC, Diago-Navarro E (2019) CRISPR-Cas influences the acquisition of antibiotic resistance in *Klebsiella pneumoniae*. PLoS ONE 14(11): e0225131. <https://doi.org/10.1371/journal.pone.0225131>

Bleriot I, Blasco L, Pacios O, Fernández-García L, López M, Ortiz-Cartagena C, Barrio-Pujante A, Fernández-Cuenca F, Pascual Á, Martínez-Martínez L, Oteo-Iglesias J, Tomás M. Proteomic Study of the Interactions between Phages and the Bacterial Host *Klebsiella pneumoniae*. Microbiol Spectr 11:e03974-22. <https://doi.org/10.1128/spectrum.03974-22>

2) Line 269: "Type IV systems are exclusively present in plasmids." Typically, the NCBI database deposits plasmids separately. In this study, were any plasmids individually analyzed? If so, please include the accession numbers.

3) Line 373-384: The authors indicated that the coexistence of both CRISPR-Cas systems and type I R-M systems improved *Klebsiella pneumoniae*'s immunity and reduced HGT, such as tetracyclin markers. In contrast, strains with both systems had significantly higher detection rates of aminoglycoside and beta-lactamase resistance genes. The authors argued that the acquisition of antibiotic resistance genes in *K. pneumoniae* is dependent not only on CRISPR arrays and type I R-M system subunits, but also on the types of antibiotic-resistant plasmids present. However, this could also apply to tetracyclin, aminoglycoside, and beta-lactamase. In my view, this explanation is insufficient. These findings suggest the possibility of additional regulatory circuits that aid this bacterium in adapting to stress and coordinating between the two systems.

4) Line 57: From From

Good work and great efforts from authors

But the isolates from 2010 to 2022 included dogs It takes a very long time to fix the same conditions, so what are the factors and parameters that help you complete the study at the same conditions?

Response Letter

Dear editors and reviewers,

We are very grateful for your constructive comments and suggestions for our manuscript entitled “Detection of CRISPR–Cas and type I R-M systems in *Klebsiella pneumoniae* of human and animal origin and their relationship to antibiotic resistance and virulence” (**Spectrum00009-24**). Your comments are very valuable and helpful for improving our manuscript. In the following, the responses to all the comments are provided one by one. In addition, we have resubmitted a new manuscript in the revised state, with the revisions highlighted in red.

We have tried our best to make all the revisions clear, and we hope that the revised manuscript can satisfy the requirements for publication. If there are any incorrect answers or questions in the new manuscript, please do not hesitate to let us know.

Yours Sincerely,

Dr. Jing Hu

Address: Department of Nosocomial Infection Administration, Zhujiang Hospital,

Southern Medical University, Guangzhou 510280, China

Tel: 15622316318

E-mail: hjalzh@smu.edu.cn

Response to Reviewer #2' comments

Comments for the Author: But the isolates from 2010 to 2022 included dogs It takes a very long time to fix the same conditions, so what are the factors and parameters that help you complete the study at the same conditions?

Response from the authors

Thank you for your insightful question. In this study, we retrieved the *K. pneumoniae* genomes which uploaded in the last five years (2019-2023), and then we obtained manually the isolation source and isolation time from the details page of each genome. To give a clearer understanding, in the revised manuscript, the correspondent context has been modified, which can be found in **Lines 114-126** and given below:

“All the *K. pneumoniae* genomes that are annotated as ‘chromosome’ or ‘complete’ at assembly level were retrieved from National Center for Biotechnology Information (NCBI) genome database as of the last five years (2019-2023). The isolation source and isolation time were obtained manually from the details page of each genome. For genomes of repeatedly recorded strains, the one with a higher sequencing quality was taken as applicable or otherwise taken randomly. Finally, a total of 520 sequences of *K. pneumoniae* were downloaded in this study, including 325 from humans and 195 from animals (Supplementary 1 data 1). Of which, 159 human strains were isolated from 2007 to 2022, but the isolation time of the remaining 66 strains were not recorded. The 195 animals strains were isolated from 2010 to 2022, included dogs (n = 70), cats (n = 35), canis lupus familiaris (n = 19), bovines (n = 15), horses (n = 13), Pteropus poliocephalus (n = 10), chickens (n = 9), swine (n = 9).”

Response to Reviewer #3' comments

Q1. Plasmids or phages typically encode anti-restriction proteins to enhance their ability to enter a new bacterial host and circumvent the restriction and modification (RM) system. For instance, the plasmid of carbapenemase-resistant *Klebsiella pneumoniae* could encode the anti-restriction proteins ArdA, KlcA, and their homologues. Lines 173-202, 385-388, etc.: The authors did not indicate or discuss the type of plasmids or phages found in the CRISPR sequence analysis. If possible, this could have explained some of the findings, such as the prevalence of carbapenem resistance or the potential for some phage resistance mechanisms.

Response from the authors

Thank you very much for the positive feedback and constructive points. We agree with the reviewer's comment, in lines 173-202, i.e. section 3.2 of the results, we only roughly described the origin of the spacer sequences in the type I-E CRISPR-Cas system from different hosts, but did not specify what the sources were. This was an oversight on our part, and the sources of which were added in Supplementary 2. We found that there were some antibiotic resistant plasmids which were homologous to the spacer sequences, such as pKPC2_KP26, pCY-CTXM-15, pSYCC2_tmex_279k, etc., and strains carrying the CRISPR-Cas system with spacer sequences homologous to the plasmids encoding carbapenem-resistance were indeed not examined for resistance genes. In the revised manuscript, the correspondent context has been modified, which can be found in **Lines 211-214** and **Lines 335-341** and given below:

“The study clarified the plasmids and phages which were homologous to the spacer sequences of 144 CRISPR-Cas system-positive strains (Supplementary 2), and there are some antibiotic resistant plasmids such as pKPC2_KP26, pCY-CTXM-15, pSYCC2_tmex_279k.”

“Also, we found that there were some antibiotic resistant plasmids which were homologous to the spacer sequences, such as pKPC2_KP26, pCY-CTXM-15,

pSYCC2_tmex_279k, etc., and strains carrying the CRISPR-Cas system with spacer sequences homologous to the plasmids encoding carbapenem-resistance were indeed not examined for resistance genes. These indicated that the CRISPR-Cas systems in *K. pneumoniae* probably played a role in regulating HGT.”

Q2. Line 269: “Type IV systems are exclusively present in plasmids.” Typically, the NCBI database deposits plasmids separately. In this study, were any plasmids individually analyzed? If so, please include the accession numbers.

Response from the authors

Thank you for your kind suggestion. We agree with the reviewer that more data would be useful to understand type IV CRISPR-Cas system. This study downloaded sequences of *K. pneumoniae* strains annotated to “chromosome” or “complete”, excluding plasmids. As the reviewer suggested, we downloaded the plasmids of all relevant *K. pneumoniae* strains from the NCBI database according to the whole genome sequence numbers of the available strains, and the accession numbers were shown in Supplementary 1 data 1. Type IV CRISPR-Cas system was not found on any of these plasmids. We also did not find typical resistant plasmids among the strains carrying the type IV-A3 CRISPR-Cas system, such as the IncFII plasmid that caused pandemic transmission of ST11 *K. pneumoniae*. In the revised manuscript, the corresponding explanation have been added, which can be found in **Line 281-282** and given below:

“In our study, the type IV-A CRISPR-Cas systems were found in chromosomes.”

Q3. Line 373-384: The authors indicated that the coexistence of both CRISPR-Cas systems and type I R-M systems improved *Klebsiella pneumoniae*'s immunity and reduced HGT, such as tetracyclin markers. In contrast, strains with both systems had significantly higher detection rates of aminoglycoside and beta-lactamase resistance genes. The authors argued that the acquisition of antibiotic resistance genes in *K.*

pneumoniae is dependent not only on CRISPR arrays and type I R-M system subunits, but also on the types of antibiotic-resistant plasmids present. However, this could also apply to tetracyclin, aminoglycoside, and beta-lactamase. In my view, this explanation is insufficient. These findings suggest the possibility of additional regulatory circuits that aid this bacterium in adapting to stress and coordinating between the two systems.

Response from the authors

Thank you for the highly valuable comment. It is true that we did not take into account the impact of other regulatory pathways that aid this bacterium in adapting to stress and coordinating between the two systems. The relationship between the two systems and the different drug-resistant plasmids is intricate and only a statistical analysis was done in this study. Further study is needed to verify the specific impact mechanisms. In the revised manuscript, the corresponding explanation and conclusion have been added, which can be found in **Lines 410-413** and given below:

“We supposed that the efficiency of the antibiotic resistance gene acquisition depended not only on the CRISPR arrays and the subunits of type I R-M systems in *K. pneumoniae* but also on the types of antibiotic-resistant plasmids, and there is the possibility of additional regulatory circuits that aid this bacterium in adapting to stress and coordinating between the two systems. Further study is needed to verify the specific impact mechanisms.”

Q4. Line 57: From From

Response from the authors

Thank you for your reminder. We are sorry for our careless mistakes. As suggested by the reviewer, the repeat word “from” has been removed in the revised manuscript, as seen in **Line 55**.

Thanks to the professional comments again that point out the above problems. We

hope these explanations would answer your doubts.

Re: Spectrum00009-24R1 (Detection of CRISPR–Cas and type I R-M systems in *Klebsiella pneumoniae* of human and animal origin and their relationship to antibiotic resistance and virulence)

Dear Dr. Jing Hu:

Your manuscript has been accepted, and I am forwarding it to the ASM production staff for publication. Your paper will first be checked to make sure all elements meet the technical requirements. ASM staff will contact you if anything needs to be revised before copyediting and production can begin. Otherwise, you will be notified when your proofs are ready to be viewed.

Sincerely,
Bobby Warren
Editor
Microbiology Spectrum

Reviewer #2 (Comments for the Author):

What are the condition factors that used for the isolates from 2010 to 2022?
What is the type of plasmids or phages found in sequence analysis of the CRISPR?

Reviewer #3 (Comments for the Author):

Thank you for addressing my comments.